# The Case of a Patient with Limited Systemic Sclerosis and Interstitial Lung Disease Overlapping with Systemic Lupus Erythematosus

**Karolina Krawczyk** [1], **Ewelina Mazur** [1], **Jaromir Kargol** [2], **Robert Kijowski** [1] and **Adam Reich** [1,*]

1 Department of Dermatology, Institute of Medical Sciences, Medical College of Rzeszow University, 35-055 Rzeszow, Poland; karolinakrawczyk10@wp.pl (K.K.); mazur.eveline@gmail.com (E.M.); ka.kijowska@gmail.com (R.K.)

2 Department of Radiology, Institute of Medical Sciences, Medical College of Rzeszow University, 35-055 Rzeszow, Poland; qjarq123@gmail.com

* Correspondence: adamandrzejreich@gmail.com

**Abstract:** About 20% of patients with systemic sclerosis have symptoms of another connective tissue disease (CTD). Interstitial lung disease (ILD) is one of the most common organ manifestations in systemic sclerosis (SSc) as well as viral illnesses, such as COVID-19, and can lead not only to diffuse alveolar damage, but also trigger an exacerbation of fibrosis among patients with preexisting ILD. It is also associated with substantial morbidity and mortality. According to the World Scleroderma Foundation, SSc-ILD can mask or mimic early COVID-19 lesions and there are no available computed tomography guidelines on how to discern those two conditions. We present a case of systemic sclerosis exacerbation after COVID-19 in a patient with SSc-Lupus Overlap Syndrome.

**Keywords:** systemic sclerosis; interstitial lung disease; systemic lupus erythematosus; mixed connective tissue disease; overlap syndrome; COVID-19

## 1. Introduction

About 20% of patients with systemic sclerosis (SSc) have symptoms of another connective tissue disease (CTD). They may coexist as two independent disease entities, while sometimes can demonstrate features common for both entities making the distinction hardly possible [1]. Interstitial lung disease (ILD), which can cause pulmonary fibrosis via diffuse parenchymal infiltrative processes, is one of the most common organ manifestations in SSc; however, it is also an important pulmonary manifestation of systemic lupus erythematosus (SLE) and other CTD. After introducing angiotensin-converting enzyme (ACE) inhibitors for the prevention and treatment of SSc renal crisis, ILD became the most common cause of death in SSc [2,3].

Viral illnesses (including COVID-19) can lead to diffuse alveolar damage, but also might trigger an exacerbation of fibrosis among patients with preexisting ILD. Lungs are the primary site for COVID-19 pathology and recent literature has shown diffuse alveolar damage as the most common finding along with platelet fibrin thrombo-emboli in pulmonary vessels of patients with SARS-CoV-2 infection. The typical radiological appearance of COVID-19 pneumonia includes peripheral, bilateral, ground-glass opacities, often of rounded morphology with or without consolidation or visible intralobular lines ("crazy-paving" pattern). Later in the disease, the reverse halo sign or other findings of organizing pneumonia are seen [4]. Here, we present a patient with ILD showing features of both SSc and SLE to illustrate how difficult it could be to differentiate the overlapping symptoms of CTDs and to differentiate ILD from COVID-19 pneumonia.

## 2. Case Report

In May 2020, a 55-year-old man was admitted to the Department of Dermatology in Rzeszów due to digital ulcers and inflammation of some nail folds with dystrophic nails. The skin lesions lasted for 3 months. He also suffered from Raynaud's phenomenon as well as reported significant fatigue for the previous 8 months. Additionally, a morning pain and stiffness of joints (wrists, knees, ankles, and shoulders) had been noticed for 4 years. Physical examination revealed red, dusky purple patches involving fingers, digital ulcers distal to the proximal interphalangeal joints (PIPs), and abnormal nail fold capillaries (Figure 1). Both, "SLE-type" capillaroscopic pattern (Figure 2A) as well as "scleroderma-like" capillaroscopic pattern (Figure 2B) were noted in the capillaroscopy in various nail folds.

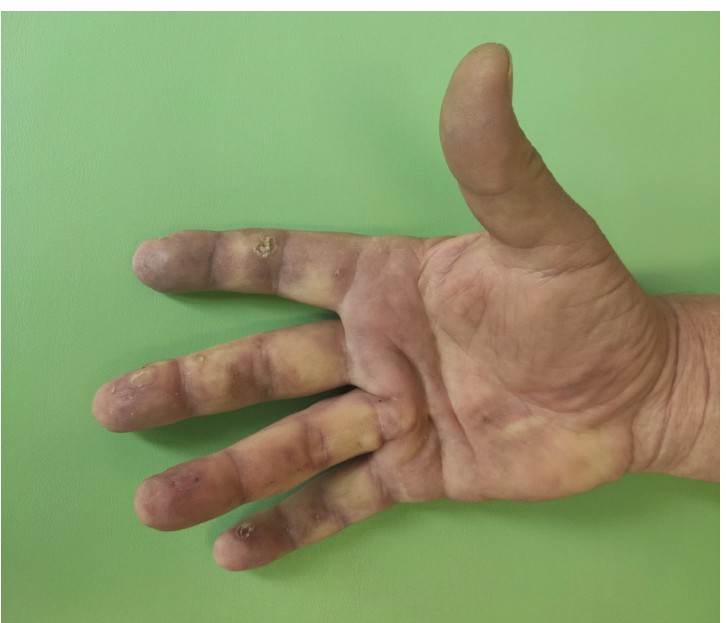

**Figure 1.** Dusky purple patches involving fingers with digital ulcers.

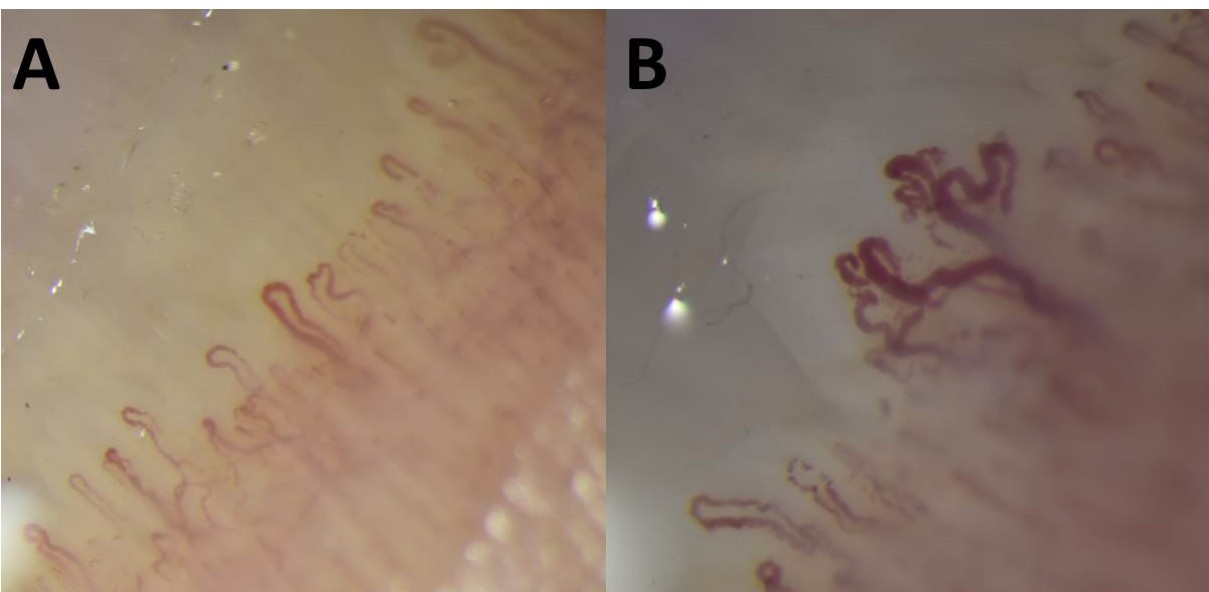

**Figure 2.** Capillaroscopy of the nail folds: numerous dilated and elongated capillaries (**A**) and giant abnormal vessels (**B**).

Laboratory tests revealed a high titer (1:1280) of antinuclear antibodies (ANA) with speckled patterns on HEp-2 cells. They were identified as SS-A and SS-B antibodies. A decreased serum level of C4 complement component (0.08 g/L, normal range: 0.1–0.4 g/L) and increase of total IgG immunoglobulins concentration (24.33 g/L, normal range: 6.0–16.0 g/L) were also observed. Additionally, proteinuria (780.9 mg/24 h) and accelerated erythrocyte sedimentation rate (51 mm/1 h) were found.

High-resolution computed tomography (HRCT) scan of the chest identified early interstitial lung lesions (Figure 3). However, no significant abnormalities were found in spirometry, body plethysmography and the carbon monoxide diffusion capacity (DLCO). X-ray examination of wrists and hands did not reveal any significant abnormalities. Left ventricular hypertrophy and diastolic dysfunction was demonstrated on sonography of the heart.

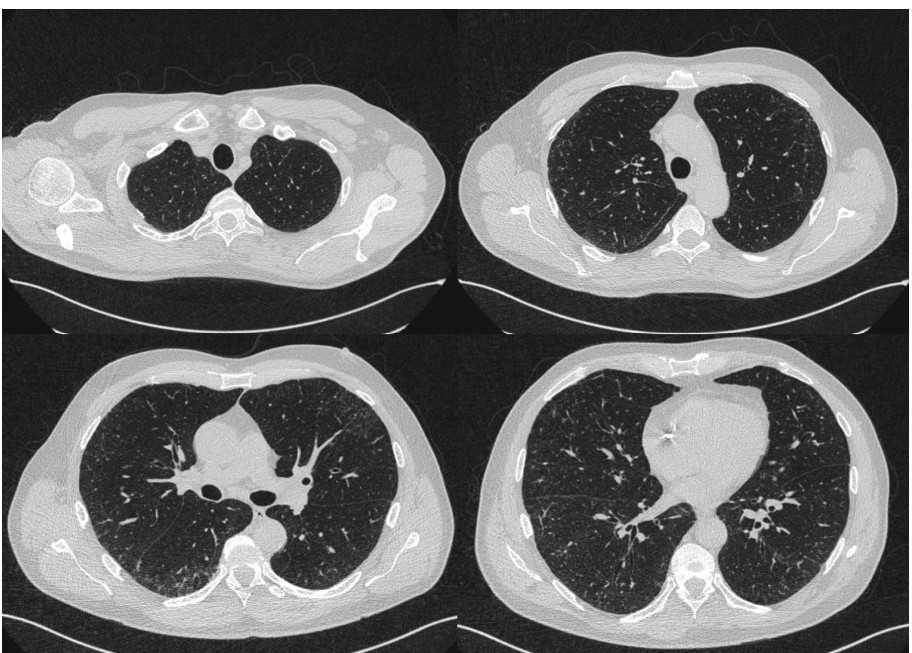

**Figure 3.** High-resolution computed tomography scan of the chest (axial, slice thickness 0.6 mm). Early bilateral interstitial lesions: thickened interlobular septa with subtle reticular pattern in basal, peripheral and dorsal (predominant) lung regions (May 2020).

Due to some features of SLE (presence of ANA, non-erosive arthritis, skin lesions mimicking chilblain lupus, and decreased complement level), treatment with chloroquine (250 mg twice a day) was implemented. Additionally, sildenafil (50 mg per day) and amlodipine (5 mg per day) led to gradual improvement of digital lesions. However, a suspicion of overlapping SLE with SSc was raised because of ILD, digital tip ulcers and abnormal nail fold capillaries. Thus, ACE inhibitor (captopril 25 mg three times a day) was introduced to protect the kidneys, and periodic monitoring of lung function parameters was recommended [3].

During a follow-up examination in September 2020, forced vital capacity (FVC%), total lung capacity (TLC%), and DLCO remained stable. However, at the end of September 2020 the patient underwent COVID-19 infection with general weakness and slight fever (up to 38.2 degrees Celsius) for several days. After infection, the patient noticed a sharp and rapid increase of fatigue and dyspnea as well as a plummet in exercise tolerance (October 2020). In January 2021, a second HRCT detected sudden progression of lung lesions with new ground glass opacities; however, the progression was observed in places of previously existing lesions. Thus, it was concluded that HRCT indicated more of an exacerbation of SSc compared to post-COVID-19 changes (Figures 4–7). In addition, a decrease of more than 10% of FVC%, TLC%, and DLCO% as well as oxygen desaturation during the 6 min

walk test were noticed (Table 1). The third HRCT performed 6 weeks later remained stable, further supporting the suggestion of ILD progression due to CTD and not due to COVID-19 pneumonia (Figure 8).

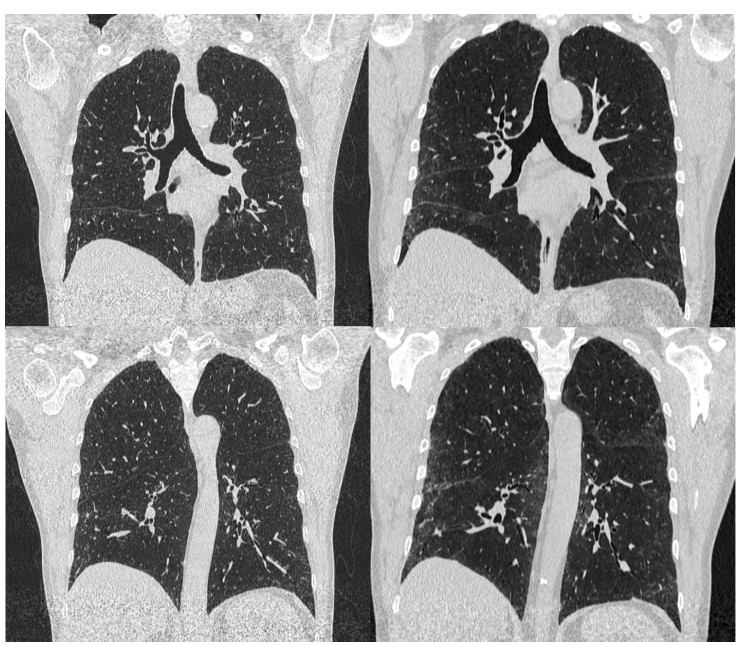

**Figure 4.** High-resolution computed tomography scan of the chest, coronal reconstruction. Comparison of lung interstitial involvement on May 2020 (**left**) and January 2021 (**right**).

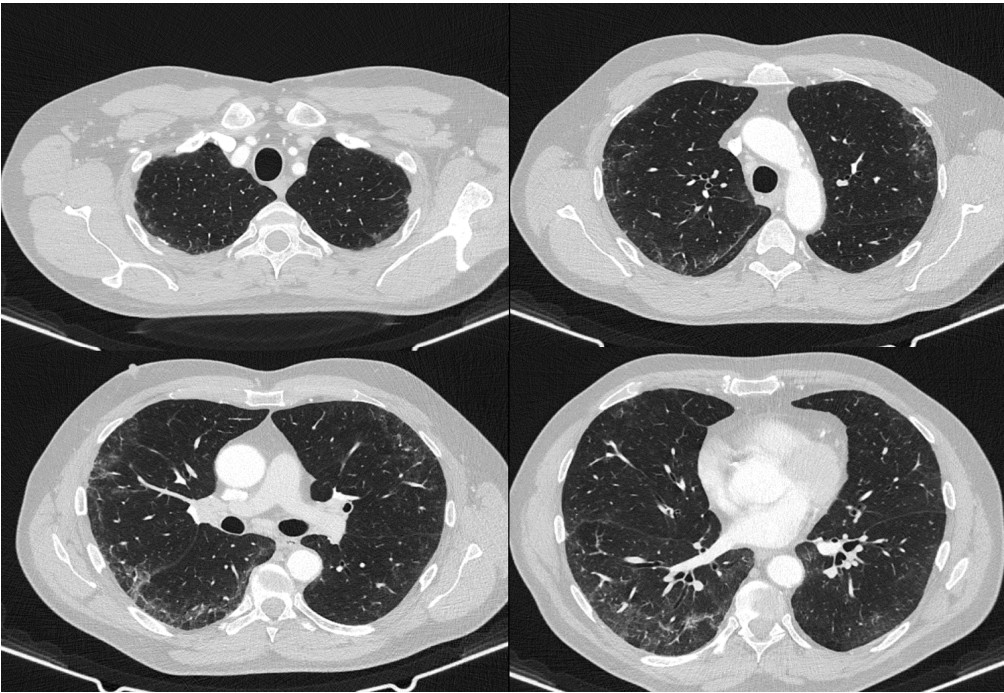

**Figure 5.** High-resolution computed tomography scan of the chest, (axial, slice thickness 1.0 mm). Thickened interlobular septa; fine reticular pattern of basal, peripheral and dorsal predominant is more prominent; ground-glass opacification superimposed on a fine reticular pattern (January 2021).

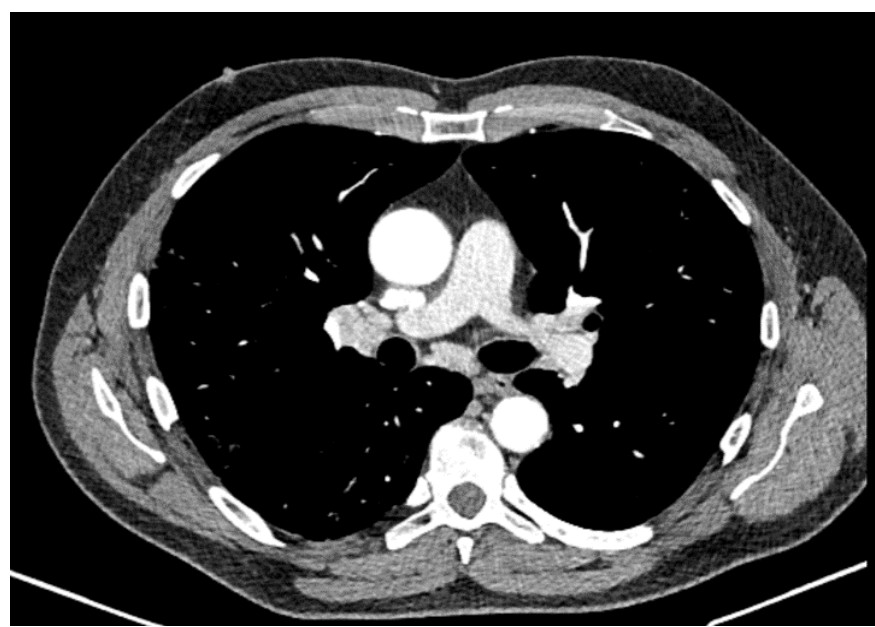

**Figure 6.** High-resolution computed tomography scan of the chest. Enlarged right (19 mm) and left hilar lymph nodes (12 mm) (January 2021).

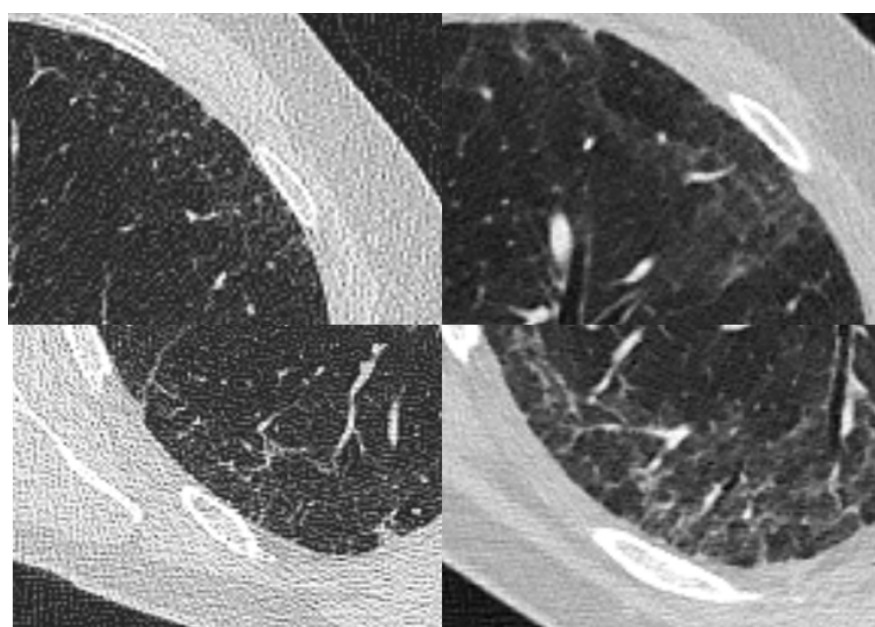

**Figure 7.** High-resolution computed tomography scan of the chest (axial, slice thickness 1.0 mm). Interstitial lesions: May 2020 (**left**)—fine reticular pattern with interlobular septa thickening. January 2021 (**right**)—ground glass opacities superimposed on a fine reticular pattern. Peripheral traction bronchiectasis.

**Table 1.** Lung function tests (DLCO—carbon monoxide diffusion capacity (>60%–<120%), FVC—forced vital capacity (80–120%), TLC—total lung capacity (80–120%)).

|       | May 2020 | September 2020 | February 2021 | Reference Values |
|-------|----------|----------------|---------------|------------------|
| FVC%  | 102.8    | 105            | 88.5          | 80–120           |
| TLC%  | 127      | 113            | 103           | 80–120           |
| DLCO% | 86       | 93             | 77            | >60–<120         |

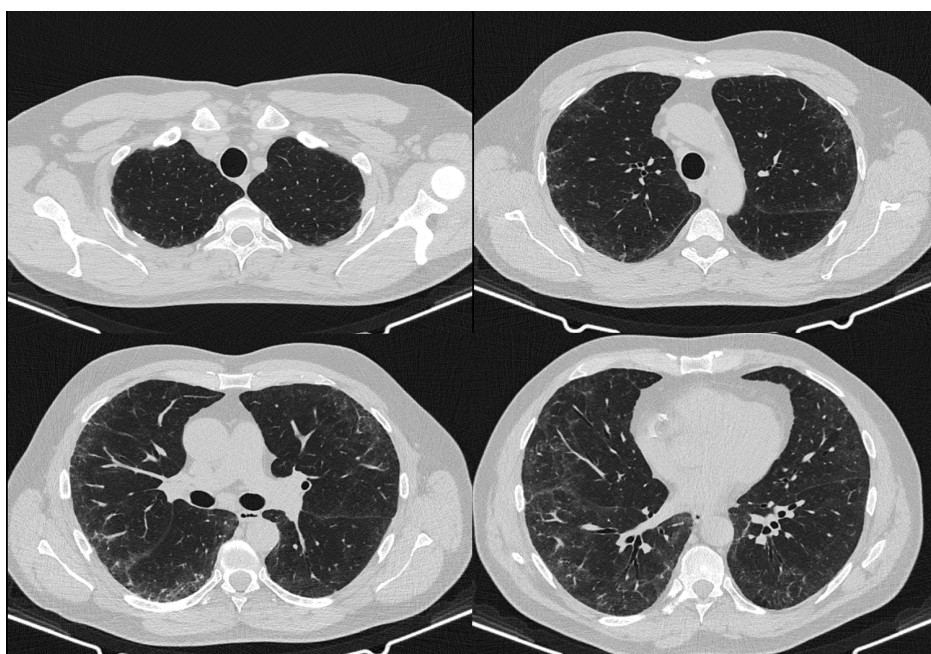

**Figure 8.** High-resolution computed tomography scan of the chest (axial, slice thickness 1.0 mm). Six weeks later the radiological image of lungs stable (February 2021).

In March 2021, the patient was admitted to our department for the next time in order to perform a routine check-out and to establish further treatment of CTD and ILD. The patient reported that he had suffered from significant dyspnea (a score of 3 on the modified Medical Research Council Dyspnea Scale [5]) since October 2020. Moreover, persistent tiredness, choke and swallowing problems were present for the previous few months. Contrast X-ray of the esophagus excluded functional disorders. At this time, diffuse swollen digits (including not only the joint capsule) was observed. Moreover, a prominent skin thickening of the dorsal region of hands and feet was noticed (a score of 9 according to the modified Rodnan skin score) [6].

Laboratory tests showed abnormal results similar to that at the disease onset, however, proteinuria declined considerably. Titer of ANA increased (1:5120) with a new nucleolar pattern on HEp-2 cells identified as RNP/Sm. Moreover, an increased level of anti-B2GP1 antibodies was observed, while U1-RNP antibody testing yielded a borderline result. Therefore, it excluded the diagnosis of mixed connective tissue disease. Based on a clinical picture and the results of additional tests, the diagnosis of limited SSc-ILD overlapping with SLE was finally made. At this point, after COVID-19 infection, the clinical symptoms of SSc were more noticeable (Table 2).

**Table 2.** Criteria of systemic sclerosis and systemic lupus erythematosus fulfilled by the presented patient.

| EULAR ACR 2013 Classification Criteria for Systemic Sclerosis (For a Definite Classification a Total Score Greater than or Equal to 9 Is Needed) | EULAR ACR 2018 Classification Criteria for Systemic Lupus Erythematosus (For a Definite Classification a Total Score of 10 or More Is Needed if Entry Criterion Has Been Fulfilled) |
| --- | --- |
| Digital tip ulcers | Antinuclear antibodies at a titer of $\geq$1:80 on HEp-2 cells |
| Abnormal nailfold capillaries | Chilblain lupus erythematosus |
| Puffy fingers | Proteinuria > 0.5 g/24 h |
| Interstitial lung disease (ILD) | Anti-B2GP1 antibodies |
| Raynaud's phenomenon | Low C4 |
| The patient has a total score of 11 | The patient has a total score of 13 and fulfills the entry criterion |

Due to deterioration of ILD, treatment with mycophenolate mofetil was implemented. The use of existing drugs was maintained. Due to the high SCORE index (7% of 10 year risk of fatal cardiovascular diseases), the patient was advised to modify the cardiovascular risk factors. Moreover, the patient started the diagnostics of pulmonary hypertension in the cardiology clinic. Currently, he is waiting for cardiac catheterization and coronary angiography.

## 3. Discussion

Lung involvement in CTDs is associated with substantial morbidity and mortality. ILD is present in approximately 40% of patients with CTD [7]. Multidisciplinary evaluation is needed to optimize the diagnostic process and management strategies. HRCT is essential in the evaluation of any suspected ILD. There is a good correlation between the degree of pulmonary impairment measured by FVC and DLCO and the extent of ILD assessed by HRCT [8]. The diagnostic approach highly relies on images of the lungs generated from volumetric scanning of the chest. HRCT allows detecting all of the abnormalities, even if subtle or focal, and to analyze lesion characteristics and distribution. In addition to the pattern of ILD, HRCT provides information on the airways, pulmonary vessels, pleura, coexisting emphysema, or cancer [9].

Patients with CTD may present various thoracic imaging abnormalities. Assessment of the presence and pattern of ILD is important and it is typically approached with the American Thoracic Society (ATS) consensus criteria for interstitial lung pneumonias [10]. Both, nonspecific interstitial pneumonia (NSIP) and usual interstitial pneumonia (UIP) pattern on HRCT of the chest can be seen in patients with CTD. Parenchymal abnormalities seen on HRCT in patients with SSc most commonly reflect an NSIP pattern, while less commonly, a UIP pattern is seen [11]. Radiologic diagnosis usually does not require histopathologic confirmation, unless unexpected imaging findings or atypical clinical features do suggest the necessity of further evaluation. UIP pattern on HRCT is mainly characterized by peripheral and basal predominant reticulation associated with fibrotic features: architectural distortion, traction bronchiectasis, and possibly honeycombing. The NSIP pattern consists of bilateral, basal-predominant ground-glass opacity and reticulation associated with traction bronchiectasis. The presence of fibrotic abnormalities varies from minimal to pronounced, possibly with honeycombing resembling UIP [12].

The typical radiological appearance of COVID-19 pneumonia includes peripheral, bilateral, ground-glass opacities, often of rounded morphology, with or without consolidation or visible intralobular lines ("crazy-paving" pattern). Later in the disease, the reversed halo sign or other findings of organizing pneumonia are seen [5]. Although some clinical signs may be seen both in COVID-19 pneumonia and interstitial pneumonia, clinical appearance and no evident changes in 6 weeks between the latter two HRCT suggest CTD-related origin of pulmonary lesions. However, according to the European Scleroderma Trials and Research group (EUSTAR), SSc-ILD can mask or mimic early COVID-19 lesions and there are no available computed tomography guidelines on how to discern those two conditions [13].

ILD, which can cause pulmonary fibrosis via diffuse parenchymal infiltrative processes, is the most common organ manifestation in SSc, especially affecting patients with diffuse cutaneous subtype of SSc [14]. ILD accounts for about 35% and 52% of the SSc patients in Europe and North America, respectively. A study by Bergamasco et al. estimated SSc-related ILD prevalence in Europe at 1.7–4.2 and annual incidence at 0.1–0.4 per 100,000 individuals and observed that it was higher among women and older people [15]. Patients with SSc-ILD have highly variable disease courses. Available data have demonstrated that while some patients with SSc-ILD present a rapid deterioration in lung function, other patients experience a slow or even no decline over time [16,17]. According to multicenter cohort studies and available databases, main pulmonary disease amounts to 20.6–58.3% death causes among patients with SSc [18–21]. Therefore, it is essential to both diagnose

SSc-related ILD early as well as to assess its severity and identify patients prone to fast progression.

Despite there being a lack of uniform definition of SSc-ILD disease progression, many experts believe that a reproducible decline in the measured FVC relative to a baseline of ≥10%, FVC decline of 5–9% in association with a relative decline in diffusing capacity of carbon monoxide (DLCO) of ≥15% as well as changes in the radiographic extent of ILD on HRCT imaging of the chest represent progression. The literature suggests that there are four phenotypes of SSc-related ILD: (1) rapid progressors (≥10% of FVC decline or 5–9% decline in association with ≥15% DLCO decline and increased extent in reticulations in HRCT within 1–2 years); (2) gradual progressors (same as above, but the time of progression is over 2 years); (3) stabilizers (FVC decline <5% or FVC improve <5%, no change in HRCT); and (4) improvers (FVC increase >5%, decreased extent of reticulations in HRCT) [22–24]. Patients with phenotypes 1 and 2 of SSc-ILD not only require early therapeutic intervention, but also closer clinical monitoring. Factors associated with poor outcomes and accelerated rate of decline in SSc-ILD patients are positive anti-topoisomerase I antibody (anti-Scl-70) [25], anti-U11/U12 RNP [26], and candidate biomarkers, such as interleukin (IL)-6 [27], C-reactive protein (CRP) [28], CCL-18, CXCL4, and KL-6 [29–34]. Increased risk of pulmonary fibrosis progression in patients with ILD was also observed after the Epstein–Barr virus, cytomegalovirus, human herpes 8 virus, adenovirus, hepatitis C Virus, Torque–Teno virus, human immunodeficiency virus, severe acute respiratory syndrome, Middle East respiratory syndrome, and severe acute respiratory syndrome coronavirus 2 (SARS-CoV2) infections [35].

COVID-19 diagnosis at an early stage in patients with ILD may be problematic because patients normally present with cough and breathlessness in the course of its primary condition. Moreover, its pyretic response may be blunted as a result of immunosuppressive therapy. Viral illnesses (COVID-19 included) might trigger an exacerbation of fibrosis among patients with ILD. Therefore, nowadays, a high index of suspicion should be maintained for COVID-19 in patients with ILD who present with fever, deterioration in their respiratory symptoms, or a drop in oxygen saturation [36]. In addition, it has been suggested that differential diagnosis of patients with ILD and an exacerbation of unknown etiology, even with an initial negative result on RT-PCR, should include COVID-19 [37]. The recent study carried out in Italy among 526 patients with SSc showed that even with WHO-defined risk factors such as immunosuppression and presence of lung disease, SSc probably does not predispose to a severe COVID-19 course and thus can be easily missed [38]. However, with an increasing prevalence of pulmonary fibrosis following COVID-19, it poses a threat to patients already afflicted by ILD [39]. According to Chan et al., the major pathological characteristic in severe acute respiratory syndrome (SARS) patients is diffuse alveolar damage (DAD) caused by endothelial and alveolar direct and indirect epithelial injury [40]. During the first 7–10 days of SARS infection (acute stage), the lungs demonstrate extensive edema, hyaline membrane formation, fluid and cellular exudation, the collapse of alveoli, and desquamation of alveolar epithelial cells. Simultaneously, fibrous tissue could be discovered in alveolar spaces. At 10–14 days in SARS development (medium phase-proliferative organizing stage), the lungs show fibrous organization (time-progressing) with interstitial and airspace fibrosis, reparative fibroblastic proliferation, and type II pneumocytic hyperplasia. 2–3 weeks of SARS (late phase-fibrotic stage), the lungs present dense septal and alveolar fibrosis [40,41]. The extent of lung fibrosis is presumably positively correlated with the duration of the SARS disease. According to Rai et al., in patients with a disease duration of less than 1 week, lung fibrosis appears in 4% of patients, between 1 and 3 weeks in 24%, and with time greater than 3 weeks—in 61% of patients [42]. Long-term studies show symptoms persisting in SARS patients beyond the early convalescent period in addition to imaging abnormalities [43]. A study by Zhang et al. showed that the pulmonary function of SARS survivors rapidly decreased in the first year after infection, plateaued, and remained steady for the following 12 years [44].

Data about the association between COVID-19 and SLE [45–47] or SSc [13,48,49] are scarce. However, there have been reports in the literature of COVID-19 shortly preceding a de novo diagnosis of SLE, presenting concomitantly, mimicking or worsening SLE manifestations, raising the possibility of SARS-CoV-2 being an autoimmunity trigger [50–55]. There are a few critical points of convergence between SLE and COVID-19 and they are: (1) type I interferons (IFNs) (which play a crucial role in the course of disease and outcomes of COVID-19, while patients with SLE have elevated circulating type I IFN or show over-expression of type I IFN genes in circulating immune cells); (2) neutrophil extracellular traps (NETs= web-like chromatin fibers with microbicidal proteins enhance inflammation and type I IFN responses in SLE, potentiate thrombosis in antiphospholipid syndrome and recent studies have suggested its key role in COVID-19); (3) complement system (dysregulation is a classic feature of SLE, and hypocomplementemia is a marker of disease activity; complement activation has been also associated with the excessive inflammatory response seen in patients with severe COVID-19); and (4) mechanistic (mammalian) target of rapamycin (mTOR—regulates T-cell and macrophage differentiation, and is thought to play a critical role in the pathogenesis of autoimmune and inflammatory diseases, such as SLE; mTOR signaling has been shown to be decreased in pDCs from COVID-19 patients, which may negatively impact the host antiviral response [56]). According to some authors SARS-CoV-2 could act as a predisposing factor for the development of a rapid autoimmune and/or autoinflammatory dysregulation, leading to severe interstitial pneumonia, in genetic predisposed individuals via the shared pathogenetic mechanisms and clinical-radiological aspects between hyper-inflammatory diseases and COVID-19 [45].

Moreover, a few articles stated that the COVID-19 pandemic could significantly increase incidence of autoimmune diseases (ADs), particularly those associated with HLA-B*08:01, HLA-A*024:02, HLA-A*11:01, and HLA-B*27:05. Populations harboring these alleles (mainly Southeast Asia, East Asia, and Oceania) are highly at risk for the associated ADs following the SARS-CoV-2 infection [57–59]. In patients with preexisting ILD, COVID-19 infection has led to its acute exacerbation (defined as subacute worsening of dyspnea and hypoxemia, new pulmonary infiltrates on imaging, and absence of pulmonary emboli, cardiac failure, and other non-pulmonary causes) [60,61]. The pathogenesis of post-viral pulmonary fibrosis is most likely due to the release of matrix metalloproteinase causing epithelial and endothelial injury; however, vascular endothelial growth factor, interleukin-6, and TNF- alpha are also being implicated [62,63].

It may be concluded that early treatment, focused on preserving lung function is of key importance for future quality of life among patients with ILD after SARS-CoV2 infection. We also speculate that COVID-19 infection might lead in some ILD cases to progressive fibrosing lung disease, but long-term follow up data are needed to confirm this suspicion.

**Author Contributions:** Conceptualization K.K., E.M., J.K. and A.R.; writing—original draft preparation K.K., E.M. and J.K.; writing—review and editing K.K., E.M., J.K., A.R. and R.K.; supervision and funding acquisition A.R. All authors have read and agreed to the published version of the manuscript.

**Funding:** This research received no external funding.

**Institutional Review Board Statement:** The study was conducted in accordance with the Declaration of Helsinki and was approved by University of Rzeszow Ethics Committee (7/11/2021, the date of approval: 19 November 2020).

**Informed Consent Statement:** Written informed consent has been obtained from the patient to publish this paper including publication of clinical photographs.

**Data Availability Statement:** No additional data were created or analyzed in this study. Data sharing is not applicable to this article.

**Conflicts of Interest:** The authors declare no conflict of interest.

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
