# Peer review of "The Case of a Patient with Limited Systemic Sclerosis and Interstitial Lung Disease Overlapping with Systemic Lupus Erythematosus"

_dermato, doi:10.3390/dermato1020009_

Round 1
Reviewer 1 Report
Authors describe a case of systemic lupus erythematosus in overlap with systemic sclerosis that develop Covid19 infection with affection of lungs (interstitial lung disease). To the authors, ILD seem not be related to Covid19 but rather to systemic sclerosis due to the progression of lesions in a time lapse of three months.
Although the authors’ attempt to describe a case of ILD development under a Covid19 active infection is of interest, the present case shows several incongruences. First of all, the SSc diagnosis is not completely convincing. It does not present minimum criteria (according for example to ACR/EULAR classification criteria of 2013) since it is not described how and if the skin is involved, patient do not present SSc specific autoantibodies, capillaroscopy is described as abnormal but it is not specified if present the typical scleroderma pattern. He has Raynaud phenomenon that can be present also in other systemic autoimmune diseases. Moreover, the digital ulcers showed in the figure are not in the fingers tips, they rather seem related to SLE digital vasculitis.
Patient seem to present more a SLE diagnosis (SSA and SSB, digital vasculitis, low C4, renal involvement, Raynaud, later development of RNP/Sm positivity, Anti B2GP1 pos) with ILD. It is true that the prevalence of ILD in SLE is lower than other systemic autoimmune diseases but still about a 9% of SLE patients can show it. Moreover, ILD could have been directly related to SSA antibodies or due to Covid19 infection.
I suggest to the authors to check again the case, see if patient meets minimum SSc criteria or not and to write again their case report.
Discussion and bibliography have also to be edited, (reference number 1 repeated twice).
Author Response
We are grateful to the reviewer for his highly valuable comments. We agree with the reviewer that the patient is unclear, but in our opinion, features of both, SLE and systemic sclerosis were present in our patient. We have also provided table 2 to summarize all features of described diseases. We have also corrected the reference list.
Reviewer 2 Report
Authors present a case of systemic sclerosis exacerbation after COVID-19 in a patient with SSc-Lupus Overlap Syndrome. In this case report, the authors say that it is difficult to distinguish COVID-19-associated pneumonia from CTD-ILD. Early detection and treatment is important because both of these diseases can leave damage to the lungs and lead to a decline in lung function. However, this case makes us think about the difficulty of diagnosis and treatment, because the treatment plan depends on which type of pneumonia it is.
I have some questions.
1) Is it possible that the symptoms worsened with the natural course of SSc?
2) There have been many reports on ILD and viral infections, but what was the impact of this COVID-19 morbidity on other manifestations of SSc (vascular disorders, intestinal symptoms, fibrosis of skin, etc.)? Only lung?
3) What are the possible mechanisms of SSc-ILD exacerbation, if you think this worsening is due to COVID-19?
4) Why did proteinuria disappear after COVID-19?
Author Response
We are grateful to the reviewer for the very positive comments. We have modified the discussion and provided more details about the role of COVID19 in the pathogenesis of SLE and systemic sclerosis.
Reviewer 3 Report
Extremely important observation in a case report.
The findings are useful and this is one of the few times when more cases to follow. Future studies can be designed to further elucidate this clinical condition as the Covid19 pandemic remains with us for the foreseeable future. The English language use is good. The parts of the results and discussion are appropriate. The article could use a table or a list of tasks laboratory tests to exclude other acute clinical conditions interfering with the clinical picture
Author Response
We are grateful to the reviewer for his very supportive comments. We have modified the manuscript according to his comments. We hope that the revision will be accepted for publication.
Round 2
Reviewer 1 Report
Discussion section: about reference 13, authors talk about world scleroderma foundation (that is a patient foundation). Reference 13 referrs to a study of EUSTAR The European Scleroderma Trials and Research group (EUSTAR).
References are still wrong in the text. See reference 38 and 39 in discussion section for example. And so on.
Authors say in the discussion "Moreover, a few articles stated that pandemic COVID-19 and the vaccination against this pathogen could significantly increase incidence of autoimmune diseases (ADs), particularly those associated with HLA-B*08:01, HLA-A*024:02, HLA-A*11:01, and HLA-B*27:05. Populations harboring these alleles (mainly Southeast Asia, East Asia,
and Oceania) are highly at risk for the associated ADs following the SARS-CoV-2 infection or vaccination [57-59]. "
I would not emphatize that vaccination is a risk factor for AD occurence since these data are not clear in litterature. Moreover the articles cited by the authors in support to this thesis do not say this concept with experimental demostrations. One article says even the opposite (PMID: 34134555) and one article just put this as a possibility (PMID: 34447375). Authors should delete or underly that there is not clear evidence about vaccination and risk for AD appereance.
Author Response
We are grateful to the reviewer for a very detailed and critical assessment of our report. We have double-checked all references and corrected them if necessary. We have also changed the World Scleroderma Foundation to EUSTAR as well as we have deleted unnecessary speculation about COVID-19 vaccination. We hope, that the revised manuscript will be found suitable to be accepted for publication in Dermato.
Sincerely yours,
Adam Reich,
Department of Dermatology, Rzeszów, Poland
Round 3
Reviewer 1 Report
Authors edited the draft accordingly to suggestions. No other correction are needed